# TOWARDS ARCHITECTURE-INSENSITIVE UNTRAINED NETWORK PRIORS FOR ACCELERATED MRI

## ABSTRACT

Untrained neural networks pioneered by Deep Image Prior have recently enabled MRI reconstruction without requiring fully-sampled measurements for training. Their success is widely attributed to the implicit regularization induced by suitable network architectures. However, the lack of understanding of such architectural priors results in superfluous design choices and sub-optimal outcomes. This work aims to simplify the architectural design decisions for DIP-MRI to facilitate its practical deployment. We observe that *certain* architectural components are more prone to causing overfitting regardless of the number of parameters, incurring severe reconstruction artifacts by hindering accurate extrapolation on the un-acquired measurements. We interpret this phenomenon from a frequency perspective and find that the architectural characteristics *favoring low frequencies*, i.e., deep and narrow with unlearnt upsampling, can lead to enhanced generalization and hence better reconstruction. Building on this insight, we propose two architecture-agnostic remedies: one to constrain the frequency range of the white-noise input and the other to penalize the Lipschitz constants of the network. We demonstrate that even with just *one extra line of code on the input*, the performance gap between the ill-designed models and the high-performing ones can be closed. These results signify that for the first time, architectural biases on untrained MRI reconstruction can be mitigated without architectural modifications.

## 1 INTRODUCTION

Magnetic resonance imaging (MRI) is a mainstream imaging tool for medical diagnosis. Reconstructing MR images from raw measurements refers to the transformation from Fourier spectrum of the object in $k$-space to image space. Since acquiring full $k$-space measurements is time-consuming, under-sampled $k$-space data are often collected to reduce scan times. Accelerated MRI is thus known as an ill-posed inverse problem that conventionally requires handcrafted priors (Lustig et al., 2007; Lingala et al., 2011) to mitigate the resulting aliasing artifacts in the output images. While supervised learning methods based on convolutional neural networks (CNNs) demonstrate better reconstruction quality with fewer measurements, their training relies on paired under-sampled and fully-sampled measurements, which are expensive to acquire and raise issues on robustness and generalization when the acquisition protocol or anatomy changes (Knoll et al., 2019; 2020a).

Instead of requiring large-scale datasets for capturing prior statistics, untrained networks pioneered by deep image prior (DIP) (Ulyanov et al., 2018) require only the corrupted image itself or partial measurements and regularize the reconstruction solely through its architecture. Concretely, DIP parameterizes the unknown desired image via a neural network and optimizes the network parameters such that the output image transformed by the degradation matrix matches the actual measurements. This parameterization offers high impedance to noise and corruption, which acts as a form of implicit regularization. Studies have attributed this property to CNN's inherent spectral bias – the tendency to fit the low-frequency image signals before the high-frequency signals (Shi et al., 2021; Chakrabarty & Maji, 2019), where the choice of network architecture is shown to be critically relevant (Chakrabarty & Maji, 2019; Liu et al., 2023; Arican et al., 2022).

A number of different architectures have been employed for untrained network priors. Compared to the original encoder-decoder structure (Ulyanov et al., 2018), Deep Decoder, an under-parameterized decoder-only network with $1 \times 1$ convolutions, is shown to represent images more concisely and

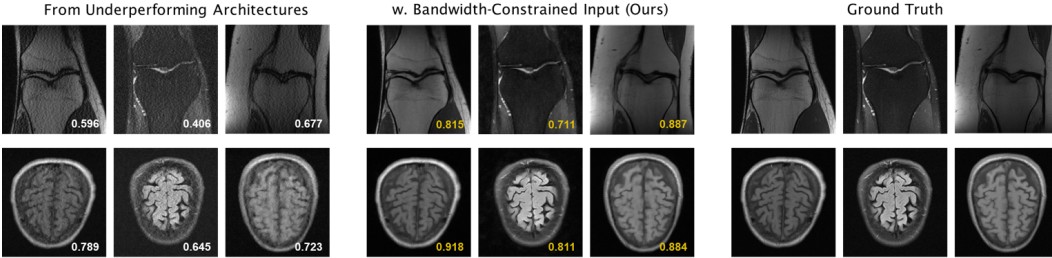

Figure 1: **Example results from underperforming architectures** with $4\times$ under-sampling. Turning the left to the right simply by low-pass filtering the white-noise input via a Gaussian blur kernel, which can be implemented as few as *one or two* lines of code. SSIM ($\uparrow$) values are reported.

employed for denoising (Heckel & Hand, 2018). Its convolutional variant is employed for MRI reconstruction and shows higher accuracy (Darestani & Heckel, 2021). Other representative untrained networks include the transformer-based networks (Korkmaz et al., 2022) and the networks discovered by neural architecture search (NAS) (Arican et al., 2022; Chen et al., 2020; Ho et al., 2021). These network architectures vary substantially in terms of the number of parameters and topology. Yet, there is generally a lack of consensus on architectural choices for a specific task. Particularly, it remains unclear what kind of architectural prior is desired in medical imaging, making it challenging to find an appropriate architecture.

In this work, we address the gap in studying the architectural influences of untrained networks in the context of accelerated MRI and demonstrate that *__the influences can be minimized in an architecture-agnostic manner__*. Without loss of generality, we focus on the typical design choices including depth, width, cross-level skip connections, upsampling types and kernel sizes. Our investigation confirms that the reconstruction outcome is sensitive to most of these basic architectural properties (Fig.1 Left). In particular, we find that underperforming architectures tend to over-fit more easily, manifested in an almost perfect fit to the available measurements but insufficient generalization to unacquired data; this issue can mainly be attributable to *certain* architectural traits that lead to faster convergence for high-frequency components, rather than the simple parameter count (Sec. 4).

Motivated by this analysis, we propose two efficient yet effective remedies, both with minimal architectural modifications: (i) constraining the effective frequency bandwidth of the white-noise input via low-pass filtering, and (ii) enforcing function smoothness via Lipschitz regularization. These techniques effectively alleviate the over-fitting issue of ill-designed architectures at little computational cost. They consistently improve the baseline models across various architectural configurations, greatly reducing the need for extensive architectural tuning. More excitingly, the architecture-agnostic nature of our methods leads to enhanced reconstruction efficiency: a **smaller**, previously under-performing network can now achieve performance on par with or even surpass that of a **larger**, heavily parameterized high-performing network.

Our contributions are three-fold:

- We provide systematic analysis on architectural sensitivity of untrained MRI reconstruction, identifying the core architectural components that critically affect the outcome and revealing the characteristics of well- and under-performing architectures.
- We alleviate the overfitting issue prevalent in under-performing architectures from a frequency perspective and propose two efficient architecture-agnostic remedies.
- Extensive experiments demonstrate that the proposed methods effectively minimize the influences due to architectural differences without requiring architectural modifications.

## 2 RELATED WORK

**Function Smoothness, Spectral Bias and Generalization.** Function smoothness, also referred to as function frequency, quantifies how much the output of a function varies with changes in its input (Fridovich-Keil et al., 2022). Spectral bias (Rahaman et al., 2019; Xu et al., 2019) is an implicit

bias that favors learning functions changing at a slow rate (low-frequency), e.g., functions with a small Lipschitz constant. In visual domains, this is evident in the network's output lacking subtle details. Many regularization techniques shown to aid generalization encourage smoothness implicitly, such as early stopping, $\ell_2$ regularizer (Rosca et al., 2020). Smoothness has been widely used as a model complexity measure in place of model size to account for the well-known "double-descent" phenomenon associated with generalization (Nakkiran et al., 2021).

To explicitly promote smoothness, a natural way is to penalize the norm of the input-output Jacobian (Novak et al., 2018; Hoffman et al., 2019). However, due to the high dimensionality of the output such as in accelerated MRI, computation of the Jacobian matrix during training is often intractable. Another efficient and prevalent solution is to constrain the network to be $c$-Lipschitz with a *pre-defined* Lipschitz constant $c$ (Miyato et al., 2018; Gouk et al., 2021). We develop a suitable form of Lipschitz regularization for untrained networks by instead penalizing *learned* Lipschitz constants, with a novel aim of achieving architecture-insensitive untrained medical image reconstruction.

**Input Frequency and Generalization.** Input has played an important role in helping neural networks represent signals of various frequencies. As in neural radiance field (NeRF) (Mildenhall et al., 2021) where coordinates are mapped to RGB values, naively training with raw coordinates as inputs results in over-smoothing; encoding the input coordinates with sinusoidal functions of higher frequencies enables the network to represent higher frequencies (Mildenhall et al., 2021; Tancik et al., 2020). Rahaman et al. (2019) also shows theoretically and empirically that fitting becomes easier for the network when the input itself contains high-frequency components. However, it has recently been reported that the high-frequency positional input encodings lead to failure of NeRF in few-shot settings due to over-fitting (Yang et al., 2023). Here, we show that this issue also applies to untrained network priors and can be addressed efficiently by the proposed methods.

**Avoid Overfitting in Untrained Networks.** Another line of efforts has been exclusively devoted to preventing overfitting to noisy images or measurements. Wang et al. (2021) propose to track the running variance of the output for an early-stopping criterion, but it is found to be unstable in medical image reconstruction (Barbano et al., 2023). Yaman et al. (2021) propose to split the available measurements into a training and a validation subset and use the latter for self-validation. However, this may result in inaccurate estimation, especially at a high under-sampling rate. Transfer-learning-based untrained networks perform pre-training on synthetic data followed by fine-tuning (Barbano et al., 2022; Nittscher et al., 2023) or subspace optimization (Barbano et al., 2023). These methods aim to use fewer trainable parameters to avoid overfitting with little performance degradation. In contrast, our methods alleviate overfitting from a frequency perspective and enable significantly better performance while maintaining the same model-wise and computation-wise complexity.

## 3 PRELIMINARIES

**Accelerated MRI** The goal of accelerated MRI reconstruction is to recover a desired image $\mathbf{x} \in \mathbb{C}^n$ ($n = n_h \times n_w$) from a set of under-sampled $k$-space measurements. We focus on a multi-coil scheme in which the measurements are obtained as:

$$\mathbf{y}_i = \mathbf{A}_i \mathbf{x} + \epsilon \text{ with } \mathbf{A}_i = \mathbf{MFS}_i, \quad i = 1, \dots, c, \tag{1}$$

where $\mathbf{y}_i \in \mathbb{C}^m$ denotes the $k$-space measurements from coil $i$, $c$ denotes the number of coils, $\mathbf{S}_i \in \mathbb{C}^n$ denotes the coil sensitivity map (CSM) that is applied to the image $\mathbf{x}$ through element-wise multiplications, $\mathbf{F} \in \mathbb{C}^{n \times n}$ denotes the 2D discrete Fourier transform, $\mathbf{M} \in \mathbb{C}^{m \times n}$ denotes the under-sampling mask, and $\epsilon \in \mathbb{C}^m$ denotes the measurement noise. Compressed sensing solves such inverse problems using Tikhonov formulation as follows (Lustig et al., 2007):

$$\mathbf{x}^* = \arg\min_{\mathbf{x}} \mathcal{L}(\mathbf{y}; \mathbf{A}\mathbf{x}) + \lambda \mathcal{R}(\mathbf{x}), \tag{2}$$

where $\mathbf{A} = \left[\mathbf{A}_1^T, \dots, \mathbf{A}_c^T\right]^T$, $\mathbf{y} = \left[\mathbf{y}_1^T, \dots, \mathbf{y}_c^T\right]^T$, and $\mathcal{L}(\bullet\,;\bullet)$ enforces consistency with the actual measurements, $\mathcal{R}(\bullet)$ is a handcrafted image regularizer (e.g., sparsity (Lustig et al., 2007).

**Untrained MRI Reconstruction** can often be framed as an **inpainting problem** where the network recovers the unacquired $k$-space measurements (masked) based on the acquired $k$-space data (observed). Compared to Eq.2, the untrained network drops the explicit image prior $\mathcal{R}(\bullet)$ and

parameterizes the image $\mathbf{x}$ via a neural network $\mathbf{G}_\theta(\mathbf{z})$ with a fixed noise input vector $\mathbf{z}$ drawn from a uniform distribution $z \sim \mathcal{U}(0,1)$:

$$\theta^* = \arg\min_\theta \mathcal{L}(\mathbf{y}; \mathbf{AG}_\theta(\mathbf{z})), \quad \mathbf{x}^* = \mathbf{G}_{\theta^*}(\mathbf{z}). \tag{3}$$

The parameterization allows novel image priors to be designed dependent on the network architecture and the associated parameters, instead of in the image space as conventionally depicted by $\mathcal{R}(\cdot)$. Nevertheless, many studies augment the untrained networks with traditional image regularizers (Liu et al., 2019), i.e., Total variation $TV(x) := \sum_{i,j} |x_{i+1,j} - x_{i,j}| + \sum_{i,j} |x_{i,j+1} - x_{i,j}|$, though it can only partially alleviate over-fitting (Nittscher et al., 2023; Barbano et al., 2023). In our experimental sections, we show that TV is not as effective as our methods or other common sparsity-inducing regularizers in improving under-performing architectures.

## 4 ARCHITECTURAL INFLUENCES

We first pinpointed the *core* architecture components that have a critical impact on the performance of untrained MRI reconstruction (*Experiment* I). Then, we investigated their relationships with other architectural properties, assessing their combined influences on the final output (*Experiment* II).

### 4.1 LOW-PASS SPATIAL KERNELS

**Experimental setup i.** Since a decoder is the minimum requirement for reconstruction, we experimented with two types of 7-layered decoder-only architectures, i.e., ConvDecoder (Darestani & Heckel, 2021) and Deep Decoder (Heckel & Hand, 2018). Experiments were performed on the 4× under-sampled multi-coil knee MRI from fastMRI database (Knoll et al., 2020b).

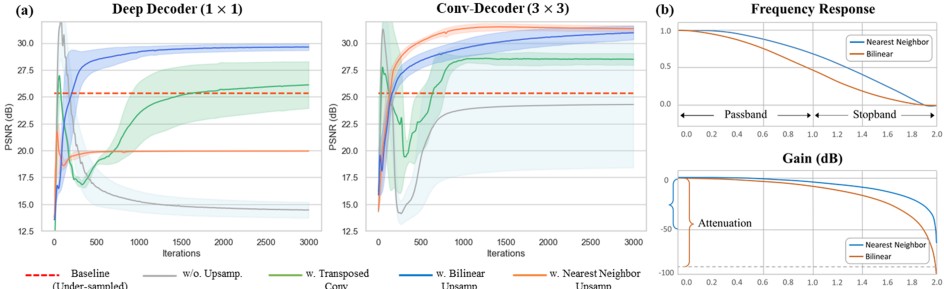

Figure 2: **Influences of architectural components**. Results averaged across three different widths.

**Upsampling.** Fig. 2(a) suggests an interesting result: removing the *unlearnt* upsampling, e.g., bilinear, leads to either failure or unstable results (see gray curves). Unlike transposed convolution, the unlearnt upsampler consists of a zero insertion step followed by a *fixed* low-pass interpolation filter that attenuates the introduced high-frequency replica and also the signal. Frequency response of bilinear interpolation filter ($\frac{\sin^2(\pi Lk)}{\pi^2 L^2 k^2}$) decays more rapidly than that of nearest neighbor ($\frac{\sin(\pi Lk)}{\pi Lk}$) as the frequency $k$ increases (Fig. 2 (b)), suggesting stronger attenuation and smoothing effects. Hence, bilinear upsampling typically biases the network towards generating smoother outputs, as prevalent in generative models (Schwarz et al., 2021). Transposed convolutions, however, are not guaranteed to be low-passed as they are *learnable*. Due to the spectral bias of network layers, they may be low-passed during early training to still enable reconstruction, but the results could be unstable (green curves).

**Kernel size.** When the unlearnt upsampling operations are absent, ConvDecoder ($3 \times 3$) still enables reconstruction while Deep Decoder ($1 \times 1$) fails completely (Fig. 2(a)). A similar phenomenon is also reported in image denoising (Chakrabarty & Maji, 2019; Liu et al., 2023). Although this may again be attributed to CNN's inherent spectral bias, the results suggest that the size of the kernel also matters, which is further corroborated by the results shown in Tab. 2, where the reconstruction accuracy becomes slightly higher as the kernel size increases to $5 \times 5$.

***Discussion***. Results of this pilot experiment suggest that the spatial kernels with low-pass characteristics, either learnable or unlearnt, are *crucial* to the success of untrained network priors. In particular, bilinear upsampling with a fixed low-pass filter produces more stable and better results (blue curves).

## 4.2 Depth, Width and Skip Connections

Here, we demonstrate that insights gained about the unlearnt upsampling can aid in understanding the connection between architectural characteristics and the reconstruction task.

**Experimental setup ii.** For this large-scale validation, we experimented with an isotropic encoder-decoder architecture used in the original DIP, i.e., equal width and kernel size for all layers throughout the network. Design choices are detailed in Tab. 1. Experiments were performed on the publicly available $4\times$ under-sampled multi-coil knee MRI from fastMRI database (Knoll et al., 2020a).

Table 1: Test bed for studying the architectural influences of an encoder-decoder untrained networks.

| Archi. Type | Depth (d) | # of Skips (s) | Width (w) | Kernel Size (k) |
|---|---|---|---|---|
| $\mathbf{A_{d-s-w-k}}$ | {2-L, 3-L, 4-L, 5-L, 8-L} | {zero, half, full} | {32, 64, 128, 256} | $\{3\times 3, 5\times 5\}$ |

Table 2: **Influences of typical architectural design choices**. *Deeper* and *Narrower* architectures tend to perform better; skip connections influence the deep architectures more; larger kernels perform slightly better. $\mathbf{A_{8-full-32-3}}$ performs the best (in lime); $\mathbf{A_{2-full-256-3}}$ performs the worst (in red).

| | | Archi. | PSNR | SSIM | Archi. | PSNR | SSIM | Archi. | PSNR | SSIM | Archi. | PSNR | SSIM |
|---|---|---|---|---|---|---|---|---|---|---|---|---|---|
| | | | | | Width (↓) | | | | | | | | |
| Depth (↑) | | $\mathbf{A_{2-full-256-3}}$ | 26.67 | 0.530 | $\mathbf{A_{2-full-128-3}}$ | 27.12 | 0.543 | $\mathbf{A_{2-full-64-3}}$ | 27.70 | 0.583 | $\mathbf{A_{2-full-32-3}}$ | 28.47 | 0.641 |
| | | $\mathbf{A_{3-full-256-3}}$ | 28.22 | 0.590 | $\mathbf{A_{3-full-128-3}}$ | 28.59 | 0.605 | $\mathbf{A_{3-full-64-3}}$ | 28.55 | 0.616 | $\mathbf{A_{3-full-32-3}}$ | 29.25 | 0.660 |
| | | $\mathbf{A_{4-full-256-3}}$ | 28.68 | 0.617 | $\mathbf{A_{4-full-128-3}}$ | 28.95 | 0.622 | $\mathbf{A_{4-full-64-3}}$ | 28.87 | 0.624 | $\mathbf{A_{4-full-32-3}}$ | 29.70 | 0.671 |
| | | $\mathbf{A_{5-full-256-3}}$ | 28.61 | 0.613 | $\mathbf{A_{5-full-128-3}}$ | 28.87 | 0.615 | $\mathbf{A_{5-full-64-3}}$ | 29.33 | 0.648 | $\mathbf{A_{5-full-32-3}}$ | 29.81 | 0.680 |
| | | $\mathbf{A_{8-full-256-3}}$ | 28.98 | 0.625 | $\mathbf{A_{8-full-128-3}}$ | 29.33 | 0.637 | $\mathbf{A_{8-full-64-3}}$ | 29.45 | 0.651 | $\mathbf{A_{8-full-32-3}}$ | 30.04 | 0.695 |
| | | | | | Skip Connections (−) | | | | | | Kernel Size (↑) | | |
| | | $\mathbf{A_{2-half-256-3}}$ | 26.91 | 0.535 | $\mathbf{A_{2-zero-256-3}}$ | 26.83 | 0.535 | $\mathbf{A_{2-full-256-3}}$ | 26.67 | 0.530 | $\mathbf{A_{2-full-256-5}}$ | 26.98 | 0.550 |
| | | $\mathbf{A_{4-half-256-3}}$ | 28.55 | 0.621 | $\mathbf{A_{4-zero-256-3}}$ | 27.54 | 0.697 | $\mathbf{A_{5-full-256-3}}$ | 28.61 | 0.613 | $\mathbf{A_{5-full-256-5}}$ | 28.82 | 0.624 |
| | | $\mathbf{A_{8-half-256-3}}$ | 29.12 | 0.669 | $\mathbf{A_{8-zero-256-3}}$ | 28.51 | 0.609 | $\mathbf{A_{8-full-256-3}}$ | 28.98 | 0.625 | $\mathbf{A_{8-full-256-5}}$ | 29.12 | 0.634 |

**Why deeper and narrower better** (Tab. 2). Theoretically, as the number of layers (depth) or channels (width) increases, the ability of the network to learn arbitrarily high frequencies (details, noise) is typically increased (Rahaman et al., 2019). While this is true for width, we have found that the effect on depth turns out to be attenuated by unlearnt upsampling. As evidenced in Fig. 3, deeper architectures typically generate smoother images, exhibiting a stronger preference for low-frequency information, whereas shallower counterparts, *even though they have fewer parameters*, are more susceptible to noise and overfitting (red arrows). This is more evident when comparing the *same* architectures with just different upsamplers, where the architectures with bilinear upsampling (stronger attenuation) are less prone to overfitting than the ones using nearest neighbor (NN) upsampling (cyan vs. blue). Hence, it is not merely the number of parameters but the architectural characteristics promoting low frequencies that seem to be the primary reason for the high performance. Note that all these results are only achievable when *unlearnt* upsampling is involved (see gray dashed curves).

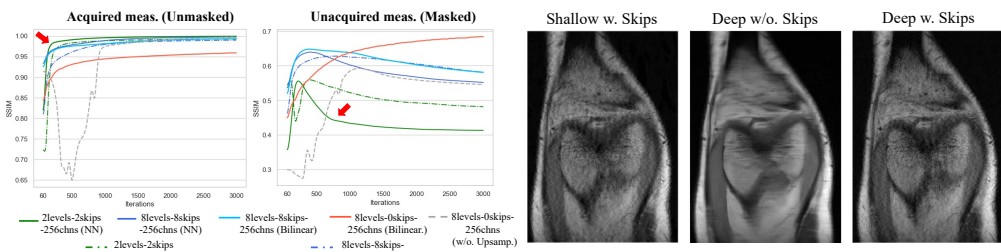

Figure 3: **Generalizability of different architectures on the masked regions**.

**Skip connections.** Deep architectures with zero skip connection converge more slowly and may lead to over-smoothing as shown in Fig. 3 (red curves). Skip connections greatly alleviate this issue and introduce more details (cyan curves), which we speculate could be due to the "reduced effective up-sampling rate". Yet, excessive skip connections make a deep architecture behave similarly as a shallower one, generating more noise (Fig. 3 right). Overall, they exert a greater influence on deeper architectures ($\mathbf{A_{8-zero}} < \mathbf{A_{8-full}} < \mathbf{A_{8-half}}$) compared to shallower ones ($\mathbf{A_{2-full}} \approx \mathbf{A_{2-zero}}$).

_**Discussion**_. Beyond the experiments here, the architecture empirically chosen for better results in such untrained inpainting-like tasks also tends to be deeper and narrower with few or no skip connections (Barbano et al., 2022; Darestani et al., 2022; Darestani & Heckel, 2021; Ulyanov et al.,

2018), instead of the typical UNet-like model with full skip connections. Certainly, infinitely deeper and narrower architectures are not always better as they may hardly represent sufficient information of the images, leading to over-smoothness. Practitioners need to find a sweet spot. To alleviate manual tuning, we introduce in the next section our methods that can promote the low-frequency bias of a given architecture, especially those shallower and wider, without architectural modifications.

## 5 METHODOLOGY

**Bandwidth-Constrained Input.** An aspect of untrained networks that can be easily overlooked is their inputs. Conventionally, the inputs are randomly sampled from $\mathcal{U}(0,1)$ and are then mapped to an image. From a frequency perspective, such white-noise input comprises all frequencies with uniform intensities. With this view in mind, we draw an analogy between untrained networks and neural radiance fields (NeRFs) which map fourier features to RGB values (Mildenhall et al., 2021). Fourier features are sinusoid functions of the input coordinates $\mathbf{p}$, i.e., $[\sin(\mathbf{p}), \cos(\mathbf{p}), ..., \sin(2^{L-1}\mathbf{p}), \cos(2^{L-1}\mathbf{p})]$, where a larger $L$ can assist the network in representing higher-frequency functions (Tancik et al., 2020) .

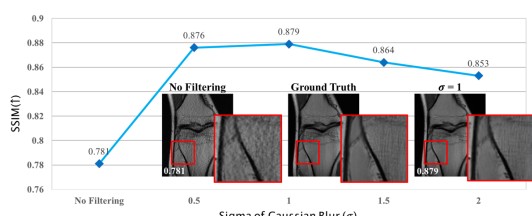

Figure 4: Limiting the frequency range of the noise input can improve underperforming architectures, $\mathbf{A_{2\text{-}full\text{-}256}}$ in this case. The output becomes smoother as $\sigma$ increases, up to a certain point.

In this sense, an untrained network can be thought of as mapping a wide range of fourier features to a target image. This enhances the network's representation ability but likely incurs over-fitting due to the faster convergence of high-frequency components. To validate this hypothesis, we applied a Gaussian blur filter $\mathcal{G}_\sigma$ on the white-noise input $z$ to remove a certain amount of high frequencies before passing it to the network, defined as $z * \mathcal{G}_\sigma$, where $*$ denotes the convolution operation. The sigma value $\sigma$ controlling the bandwidth of the filter is the only hyperparameter. As exemplified in Fig. 4, simple tuning of $\sigma$ already brings significant gains on a shallow and wide architecture without architectural changes. Similar in spirit, recent work by Yang et al. (2023) also shows that masking the high-frequency Fourier features helps NeRFs generalize in few-shot settings.

**Lipschitz Regularization.** Spectral bias towards low frequencies favors functions that do not change at a high rate, i.e., functions with small Lipschitz constants. A function $f : \mathcal{X} \to \mathcal{Y}$ is said to be Lipschitz continuous if there is a constant $k > 0$ such that

$$\|f(\mathbf{x}_1) - f(\mathbf{x}_2)\|_\mathrm{p} \leq k\|\mathbf{x}_1 - \mathbf{x}_2\|_\mathrm{p} \quad \forall \mathbf{x}_1, \mathbf{x}_2 \in \mathcal{X}, \tag{4}$$

where $k$ is the Lipschitz constant that bounds how fast $f$ can change globally w.r.t. input perturbations.

Shi et al. (2021) upper bound the Lipschitz constants of the untrained network layers to pre-defined and manually chosen values, as the optimal value may vary with tasks. Instead, we make the per-layer Lipschitz bounds learnable and regularize their magnitudes during optimization.

The Lipschitz constant of a convolutional layer is bounded by the operator norm of its weight matrix (Gouk et al., 2021). To bound a convolutional layer to a specific Lipschitz constant $k$, the layer with $m$ input channels, $c$ output channels and kernels of size $w \times h$ is first reshaped to a 2-D matrix $W^\ell \in \mathbb{R}^{n \times cwh}$, and then normalized as:

$$\tilde{W}_\ell = \frac{W_\ell}{\max(1, \frac{\|W_\ell\|_\mathrm{p}}{\mathrm{SoftPlus}(k_\ell)})}, \tag{5}$$

where $k_\ell$ is a learnable Lipschitz constant for each layer, $\|\cdot\|_\mathrm{p}$ is chosen as the $\ell_\infty$ norm and $\mathrm{SoftPlus}(c^l) = \ln(1 + \exp(k_\ell))$ ensures the learned Lipschitz bounds are non-negative as in Liu et al. (2022). Such formulation *only* normalizes $W_\ell$ if its matrix norm is larger than the learned Lipschitz constraint during training. The ultimate Lipschitz regularization is defined in a similar way as in Yoshida & Miyato (2017) but with learned Lipschitz constants:

$$\min_{\Theta, K} \mathcal{L}(\mathbf{y}; \mathbf{AG_\Theta}(\mathbf{z})) + \lambda \sum_{l=1}^{L} \mathrm{SoftPlus}(\mathbf{k}_\ell)^2 \tag{6}$$

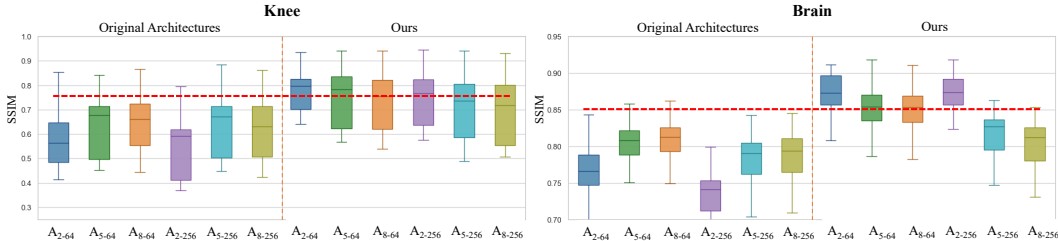

Figure 5: Our methods significantly improve architectures with various widths and depths. **A 2-level architecture can now surpass the 8-level counterparts**. - - - denotes the average median values of the results from the improved architectures. All compared architectures have full skip connections.

Table 3: Number of parameters of the compared architectures. M: million(s)

| Archi. | $\mathbf{A_{2\text{-}64}}$ | $\mathbf{A_{5\text{-}64}}$ | $\mathbf{A_{8\text{-}64}}$ | $\mathbf{A_{2\text{-}256}}$ | $\mathbf{A_{5\text{-}256}}$ | $\mathbf{A_{8\text{-}256}}$ |
|---|---|---|---|---|---|---|
| # of Param. | 0.24 M | 0.59 M | 0.95 M | 3.7 M | 9.3 M | 14.8 M |

where $\mathbf{K}$ is a collection of learnable Lipschitz constant $k_\ell$ of each layer jointly optimized with the network parameters, and $\lambda$ controls the granularity of smoothness.

## 6 EXPERIMENTS

### 6.1 DATASETS

Experiments were performed on the publicly available multi-coil knee and brain datasets of fastMRI (Knoll et al., 2020b). Data acquisition used a conventional Cartesian 2D TSE protocol with a 15-channel coil array and a matrix size of $320 \times 320$. The knee dataset includes 798 cases of proton-density weighting with fat suppression (PDFS) and 796 cases without fat suppression (PD). 50 knee slices and 50 AXT1PRE axial brain slices sampled from the knee and brain validation datasets were used for evaluating all compared methods, respectively. Ground truth was obtained by computing the root-sum-of- squares (RSS) reconstruction method applied to the fully sampled $k$-space data. The $k$-space data used for reconstruction was retrospectively masked by selecting 25 central $k$-space lines along with an uniform undersampling at outer $k$-space, achieving a total of $4\times$ acceleration.

### 6.2 IMPLEMENTATION DETAILS

The base architecture we evaluated on are N-level encoder-decoder architectures as used in the original DIP, with each level consisting of two consecutive convolutions, nearest neighbor up-sampling, ReLU activation function, batch normalization (Ioffe & Szegedy, 2015) and zero padding. Skip connections are implemented via concatenation. The architectures are isotropic with the same width and kernel size throughout the layers. All evaluated architectures are trained for 3000 iterations using mean absolute error (MAE) and Adam optimizer with a learning rate of 0.008. The results from the last iteration are reported. We additionally evaluated two commonly used regularizers for combating overfitting: TV and L2. Hyperparameters are carefully chosen for each regularization method. $\lambda$ is set to 1 for Lipschitz regularizer, 0.001 for TV regularizer, and $1e^{-5}$ for $\ell_2$ regularizers. The noise input is drawn from a uniform distribution $z \sim \mathcal{U}(0, 1)$. The filter size of the Gaussian blur was set to 3 and the sigma value was uniformly chosen from $[0.5, 2.0]$ for every slice in all the experiments. Peak signal-to-noise ratio (PSNR) and structural similarity index (SSIM) are used for quantitative evaluation. The training was conducted with one GPU (TITAN X, 12GB).

### 6.3 IMPROVING UNDER-PERFORMING ARCHITECTURES

Fig. 5 quantitatively demonstrates the substantial improvement using our methods on architectures of various widths and depths evaluated on both knee and brain datasets. As discussed in Sec. 4, originally, shallower architectures tend to perform the worst, i.e., $\mathbf{A_{2\text{-}64}}$ and $\mathbf{A_{2\text{-}256}}$. $\mathbf{A_{2\text{-}64}}$ even has the least number of parameters (Tab. 3). This again shows that the model size itself does not

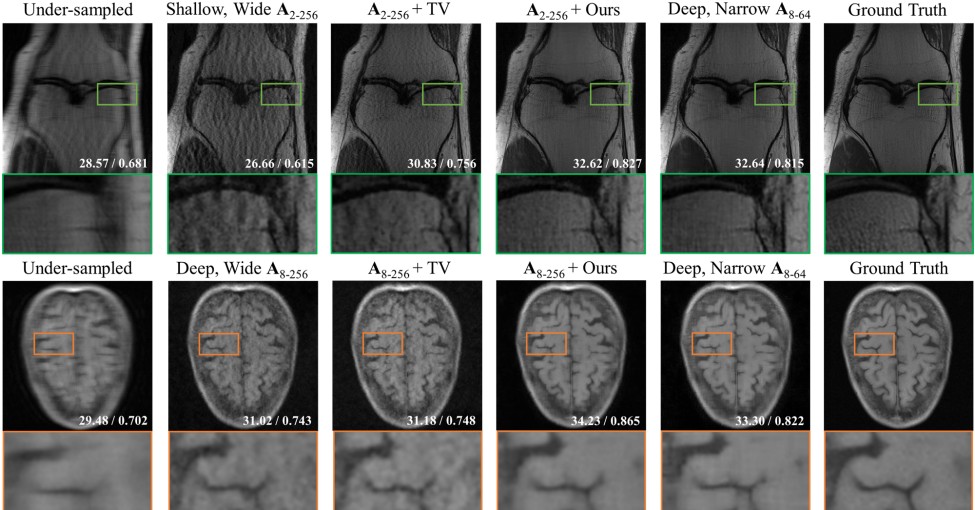

| Under-sampled | Shallow, Wide $\mathbf{A}_{2\text{-}256}$ | $\mathbf{A}_{2\text{-}256}$ + TV | $\mathbf{A}_{2\text{-}256}$ + Ours | Deep, Narrow $\mathbf{A}_{8\text{-}64}$ | Ground Truth |
|---|---|---|---|---|---|
| 28.57 / 0.681 | 26.66 / 0.615 | 30.83 / 0.756 | 32.62 / 0.827 | 32.64 / 0.815 | |
| Under-sampled | Deep, Wide $\mathbf{A}_{8\text{-}256}$ | $\mathbf{A}_{8\text{-}256}$ + TV | $\mathbf{A}_{8\text{-}256}$ + Ours | Deep, Narrow $\mathbf{A}_{8\text{-}64}$ | Ground Truth |
| 29.48 / 0.702 | 31.02 / 0.743 | 31.18 / 0.748 | 34.23 / 0.865 | 33.30 / 0.822 | |

Figure 6: Our methods enable the underperforming architectures to perform similarly to or even better than the well-performing architectures (i.e., Deep, Narrow). Full skip connections in all architectures.

Table 4: Quantitative evaluation of different regularization methods on **knee** datasets. The best and the second-best are highlighted. Results from the last iteration (3000th) are reported.

| Regularizers | $\mathbf{A}_{2\text{-}256}$ | $\mathbf{A}_{2\text{-}64}$ | $\mathbf{A}_{5\text{-}256}$ | $\mathbf{A}_{5\text{-}64}$ | $\mathbf{A}_{8\text{-}256}$ | $\mathbf{A}_{8\text{-}64}$ | $\mathbf{A}_{2\text{-}256}$ | $\mathbf{A}_{2\text{-}64}$ | $\mathbf{A}_{5\text{-}256}$ | $\mathbf{A}_{5\text{-}64}$ | $\mathbf{A}_{8\text{-}256}$ | $\mathbf{A}_{8\text{-}64}$ |
|---|---|---|---|---|---|---|---|---|---|---|---|---|
| | PSNR ↑ | | | | | | SSIM ↑ | | | | | |
| TV | 28.25 | 27.85 | 29.33 | 29.57 | 29.54 | 30.01 | 0.588 | 0.592 | 0.635 | 0.651 | 0.645 | 0.687 |
| L2 | 29.80 | 30.16 | 29.64 | 29.78 | 30.03 | 31.30 | 0.694 | 0.678 | 0.686 | 0.690 | 0.693 | 0.715 |
| Lips. Reg. (Ours) | 28.41 | 29.21 | 29.17 | 29.79 | 29.43 | 30.14 | 0.601 | 0.600 | 0.629 | 0.651 | 0.636 | 0.666 |
| Bandw.Const. Input (Ours) | 30.87 | 30.89 | 30.02 | 31.24 | 29.31 | 30.89 | 0.739 | 0.768 | 0.694 | 0.748 | 0.698 | 0.727 |
| Bandw.Const. Input + Lips. Reg. (Ours) | 31.61 | 31.93 | 29.40 | 31.67 | 29.82 | 31.58 | 0.750 | 0.776 | 0.702 | 0.727 | 0.697 | 0.732 |
| w/o. Reg. (Baseline) | 27.18 | 27.62 | 29.16 | 29.23 | 28.98 | 29.35 | 0.541 | 0.575 | 0.628 | 0.640 | 0.625 | 0.644 |

Table 5: Quantitative evaluation of different regularization methods on **brain** datasets.

| Regularizers | $\mathbf{A}_{2\text{-}256}$ | $\mathbf{A}_{2\text{-}64}$ | $\mathbf{A}_{5\text{-}256}$ | $\mathbf{A}_{5\text{-}64}$ | $\mathbf{A}_{8\text{-}256}$ | $\mathbf{A}_{8\text{-}64}$ | $\mathbf{A}_{2\text{-}256}$ | $\mathbf{A}_{2\text{-}64}$ | $\mathbf{A}_{5\text{-}256}$ | $\mathbf{A}_{5\text{-}64}$ | $\mathbf{A}_{8\text{-}256}$ | $\mathbf{A}_{8\text{-}64}$ |
|---|---|---|---|---|---|---|---|---|---|---|---|---|
| | PSNR ↑ | | | | | | SSIM ↑ | | | | | |
| TV | 29.22 | 29.61 | 31.26 | 31.37 | 31.32 | 31.64 | 0.735 | 0.764 | 0.785 | 0.802 | 0.787 | 0.807 |
| L2 | 30.51 | 31.26 | 31.64 | 32.67 | 31.61 | 32.38 | 0.805 | 0.813 | 0.811 | 0.847 | 0.819 | 0.827 |
| Lips. Reg. (Ours) | 30.92 | 29.73 | 31.47 | 32.11 | 31.50 | 32.03 | 0.795 | 0.766 | 0.792 | 0.812 | 0.800 | 0.820 |
| Bandw.Const. Input (Ours) | 33.34 | 32.67 | 32.14 | 32.66 | 32.03 | 32.92 | 0.870 | 0.866 | 0.811 | 0.849 | 0.825 | 0.849 |
| Bandw.Const. Input + Lips. Reg. (Ours) | 32.90 | 33.12 | 32.08 | 32.83 | 31.70 | 33.14 | 0.855 | 0.870 | 0.815 | 0.851 | 0.805 | 0.849 |
| w/o. Reg. (Baseline) | 29.08 | 29.41 | 31.15 | 31.42 | 31.27 | 31.68 | 0.729 | 0.761 | 0.782 | 0.801 | 0.784 | 0.807 |

fully explain the problem of overfitting. Our method brings the most significant improvements to these two types of architectures, allowing them to perform on par with or even surpass the much larger architectures, e.g., $\mathbf{A}_{8\text{-}256}$. Particularly, as shown in Tab. 4 and Tab. 5, limiting the frequency spectrum of the input *alone* already brings dramatic improvement; better results are achieved when combined with the proposed Lipschitz regularization. These results align with our hypothesis that promoting low-frequency bias for underperforming architectures could help minimize architectural influences. Fig. 6 shows qualitatively that our methods can improve a shallow and wide architecture as well as a deep architecture with full skip connections, both of which are susceptible to overfitting. Meanwhile, we also evaluated the efficacy of the commonly-used TV regularizer (Barbano et al., 2023) and L2 regularizer. Both quantitative and qualitative results demonstrate that TV is not as effective as ours or L2 in improving the underperforming architectures and alleviating overfitting. Additionally, we find that it seems to be more sensitive to the choice of hyper-parameter, which might partially explain its suboptimal performance. However, tuning the regularization granularity on a case-by-case basis is unrealistic. In contrast, **our method only requires minimal hyper-parameter tuning**. Especially, the sigma value of the Gaussian kernel is uniformly drawn at random from [0.5, 2.0] for each reconstruction. Evaluations on the sensitivity of the proposed Lipschitz regularization to choices of hyperparameters are given in Suppl.

Table 6: Quantitative comparisons with classic untrained network architectures. **A small and compact model with our modifications outcompetes the much larger models**, achieving higher efficiency.

| Datasets | | DIP | Deep Decoder | ConvDecoder | $\mathbf{A_{2\text{-}64}}$ (Plain) | $\mathbf{A_{2\text{-}64}}$ (Ours) |
|---|---|---|---|---|---|---|
| Knee | PSNR | 29.16 | 27.21 | 29.59 | 27.62 | 31.93 |
| | SSIM | 0.628 | 0.687 | 0.655 | 0.575 | 0.776 |
| Brain | PSNR | 31.23 | 26.97 | 31.81 | 29.42 | 33.12 |
| | SSIM | 0.784 | 0.747 | 0.800 | 0.761 | 0.870 |
| # of Params.(Millions) | | 9.3 M | 0.47 M | 4.1 M | 0.24 M | 0.24 M |

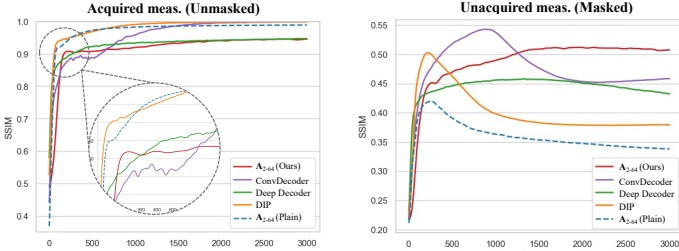

Figure 7: Most architectures more or less suffer from over-fitting. Our method significantly alleviates the overfitting issue of $\mathbf{A_{2\text{-}64}}$ and improves its extrapolation ability on the masked regions.

## 6.4 COMPARISONS WITH CLASSIC UNTRAINED NETWORK ARCHITECTURES

One major benefit of achieving architecture-insensitive reconstruction is that one may employ a smaller architecture for higher efficiency without sacrificing accuracy. Here we compared our *improved* $\mathbf{A_{2\text{-}64}}$ with other classic architectures, especially ConvDecoder, which was successfully applied to MRI reconstruction (Darestani & Heckel, 2021). Quantitative results are in Tab. 6. Qualitative results are in Suppl. Following their original setups, Deep Decoder and ConvDecoder contain 7 layers with no skip connections, but Deep Decoder does not have spatial kernels except for the unlearnt upsampling. Fig. 7 shows that compared with Deep Decoder, ConvDecoder is less prone to over-fitting and more advantageous in such inpainting-like tasks, which aligns with our analysis in Sec. 4. DIP is a 5-level hourglass architecture with full skip connections, i.e., $\mathbf{A_{5\text{-}full\text{-}256\text{-}3}}$. As expected, both DIP and the original $\mathbf{A_{2\text{-}64}}$ (with full skip connections) suffer from over-fitting, fitting the acquired measurements more readily while extrapolating poorly on the masked regions (Fig. 7). Our method greatly alleviates the overfitting issue and allows it to surpass the other models by a large margin, despite its smaller size.

## 7 CONCLUSION

This work aims to tackle the challenge of simplifying architectural decisions for untrained networks in the context of accelerated MRI reconstruction, which has been an open question. Through a series of experiments, we have identified the roles of common architectural properties – namely, depth, width, and skip connections, and uncovered that the shallower and/or wider architectures are more prone to overfitting due to the architectural configurations that cause faster convergence of high frequencies instead of the number of parameters. Importantly, this empirical evidence may unveil the profound link between frequency bias and overfitting. Based on these insights, we propose to use bandlimited inputs and Lipschitz regularization to alleviate overfitting. Both techniques are easy to implement but can dramatically improve the underperforming architectures. Besides the architectural challenge, our method could also reduce the runtime – another challenge of the untrained networks – by employing efficient compact models. Moreover, the effectiveness of our method also suggests a connection between untrained networks and NeRF, which we consider worthy of future investigation. We believe that our work not only improves the practicality of untrained networks but also contributes to a deeper understanding of their working mechanisms.

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

# A   SUPPLEMENTARY MATERIALS

## A.1   CODE AVAILABILITY

The implementation of our methods along with the datasets used will be made publicly available.

## A.2   COMPARISON TO SELF-VALIDATION

Self-validation, which uses a subset of measurements for validation, is popular in single-instance MRI reconstruction for preventing overfitting (Yaman et al., 2021; Darestani et al., 2022). It works by detecting the timing for stopping as near to the peak PSNR as possible. Our method differs in that it can generally enhance the peak PSNR of the architecture while also alleviating overfitting. This can be seen from the results of $A_{2\text{-}256}$, $A_{2\text{-}64}$, and Fig. 7. We show below that they can be combined to achieve much better performance.

Table 7: Quantitative evaluation on $4\times$ multi-coil **knee** datasets. The best and the second-best are highlighted. $5\%$ of the measurements are held out for validation. 'ws' denotes the duration (# iterations) of a sliding window that monitors the self-validation error for **automatic early stopping**.

| Methods | $A_{2\text{-}256}$ | $A_{2\text{-}64}$ | $A_{5\text{-}256}$ | $A_{5\text{-}64}$ | $A_{2\text{-}256}$ | $A_{2\text{-}64}$ | $A_{5\text{-}256}$ | $A_{5\text{-}64}$ |
|---|---|---|---|---|---|---|---|---|
| | PSNR ($\uparrow$) | | | | SSIM ($\uparrow$) | | | |
| Self-Val. (ws=30) | 29.59 | 29.59 | 31.18 | 31.05 | 0.682 | 0.695 | 0.746 | 0.744 |
| Self-Val. (ws=50) | 29.04 | 29.62 | 31.07 | 30.94 | 0.642 | 0.684 | 0.738 | 0.737 |
| Ours (3000 iters) | 31.61 | 31.93 | 29.40 | 31.67 | 0.750 | 0.776 | 0.702 | 0.727 |
| Ours w. Self-Val (ws=30) | 31.49 | 31.09 | 31.74 | 31.73 | 0.762 | 0.762 | 0.769 | 0.772 |
| Ours. w. Self-Val (ws=50) | 31.60 | 31.41 | 31.78 | 31.63 | 0.762 | 0.767 | 0.771 | 0.771 |
| Baseline (3000 iters) | 27.18 | 27.62 | 29.16 | 29.23 | 0.541 | 0.575 | 0.625 | 0.640 |

Table 8: Quantitative evaluation on $4\times$ multi-coil **brain** datasets.

| Methods | $A_{2\text{-}256}$ | $A_{2\text{-}64}$ | $A_{5\text{-}256}$ | $A_{5\text{-}64}$ | $A_{2\text{-}256}$ | $A_{2\text{-}64}$ | $A_{5\text{-}256}$ | $A_{5\text{-}64}$ |
|---|---|---|---|---|---|---|---|---|
| | PSNR $\uparrow$ | | | | SSIM $\uparrow$ | | | |
| Self-Val. (ws=30) | 30.39 | 30.06 | 32.78 | 32.48 | 0.822 | 0.832 | 0.872 | 0.868 |
| Self-Val. (ws=50) | 30.21 | 30.15 | 32.77 | 32.44 | 0.813 | 0.829 | 0.870 | 0.867 |
| Ours (3000 iters) | 32.90 | 33.12 | 32.08 | 32.83 | 0.855 | 0.870 | 0.815 | 0.851 |
| Ours w. Self-Val (ws=30) | 32.94 | 32.56 | 33.06 | 33.04 | 0.874 | 0.873 | 0.880 | 0.879 |
| Ours. w. Self-Val (ws=50) | 32.99 | 32.72 | 33.06 | 32.52 | 0.874 | 0.874 | 0.880 | 0.870 |
| Baseline (3000 iters) | 29.08 | 29.41 | 31.15 | 31.42 | 0.729 | 0.761 | 0.782 | 0.801 |

## A.3   COMPARISONS TO SUPERVISED METHODS

Supervised methods shine when test data are within the training distribution. DIP-like methods are more advantageous on out-of-distribution data as they are agnostic to changes in acquisition protocols and anatomy shift, etc.(Yaman et al., 2021). Our method accelerates DIP by allowing a more compact network to be employed, and when combined with self-validation, its runtime is further reduced.

Table 9: Robustness and runtime comparisons with U-Net on the $4\times$ multi-coil **brain** validation dataset. **In-domain** dataset: 50 AXT1PRE slices. **Out-domain** dataset: 30 AXFLAIR slices. Runtime is computed as the per-slice average for every slice of size $20 \times 640 \times 320$. The DIP $A_{2\text{-}64}$ is trained for 3000 iterations when self-validation is not used.

| | Methods | In-domain | | Out-domain | | Runtime (mean$\pm$std) | |
|---|---|---|---|---|---|---|---|
| | | PSNR | SSIM | PSNR | SSIM | Train | Inference |
| Trained | U-Net | **34.11** | **0.910** | 28.25 | 0.785 | $\geq 3$ days | $0.1 \pm 0.003$ sec |
| Untrained | $A_{2\text{-}64}$ (baseline) | 29.41 | 0.761 | 29.77 | 0.715 | – | $26.5 \pm 8.1$ mins |
| | $A_{2\text{-}64}$ (ours) | 33.12 | 0.870 | **32.45** | 0.832 | – | $26.8 \pm 8.3$ mins |
| | $A_{2\text{-}64}$ (ours) w. Self-Val. | 32.56 | 0.873 | 32.11 | **0.840** | – | $4.8 \pm 2.7$ mins |

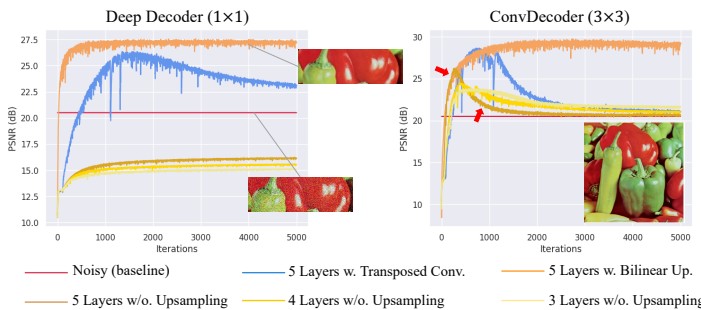

Figure 8: **Denoising** experiments. (**Left**) In non-convolutional networks, removing the upsampling hampers the denoising capability, which cannot be compensated by merely adjusting the network to be more under-parameterized. Transposed convolutions result in a more rapid decline in performance than bilinear upsampling. (**Right**) Convolutional layers *alone* exhibit certain denoising effects but necessitate early stopping. The showcased image is from the classic dataset Set9 (Dabov et al., 2007).

## A.4 THE "DEVIL" IS IN THE UPSAMPLING

Here we provide additional evidence demonstrated on brain datasets as well as natural images to support our findings about the unlearnt upsampling and its relationships with other architectural properties in DIP. These findings critically motivate our methods and lead us to conclude that the underperformance in DIP is not primarily attributed to the number of parameters.

### A.4.1 ADDITIONAL MRI EXPERIMENTS

As stated in Sec. 4.1, an *unlearnt* upsampler can be seen as a zero insertion step which increases the output sampling rate, followed by a non-ideal low-pass filter (LPF, shortened as $\mathcal{L}$) that attenuates both the introduced high-frequency replica and signals. Bilinear and nearest neighbor (NN) upsamplers differ only in the LPFs used. We additionally constructed a customized upsampler that has a greater attenuation ability than bilinear upsampling. This was done by first interleaving the feature maps of every layer with zeros and then convolving them with a handcrafted LPF: $\mathcal{L}_{-100}$, with the subscript denoting the decayed dB. The Details of construction are specified in A.7.

Table 10: **Importance of upsampling**. Evaluated on the $4\times$ multi-coil **brain** dataset. From the left to the right, the attenuation extent of the LPF increases. PSNR values at 3000th iteration are reported.

| Methods | w/o. Upsampling. | NN | Bilinear | $\mathcal{L}_{-100}$ | # of Params. (Millions) |
|---|---|---|---|---|---|
| ConvDecoder | $28.69 \pm 1.6$ | $31.78 \pm 1.2$ | $32.31 \pm 1.3$ | $32.48 \pm 1.2$ | 4.1 M |
| Deep Decoder | $24.55 \pm 1.1$ | $27.10 \pm 0.9$ | $31.36 \pm 1.4$ | $32.68 \pm 1.1$ | 0.47 M |

Tab. 10 shows that simply varying the upsampling type substantially influences the network performance such that the performance gap between the two networks can even be closed without requiring architecture scaling. Overall, the presence of unlearnt upsampling is vital to the non-convolutional Deep Decoder and enhances both the accuracy and stability of ConvDecoder: the peak PSNR is reached more slowly when the attenuation is stronger, alleviating overfitting (Fig. 9).

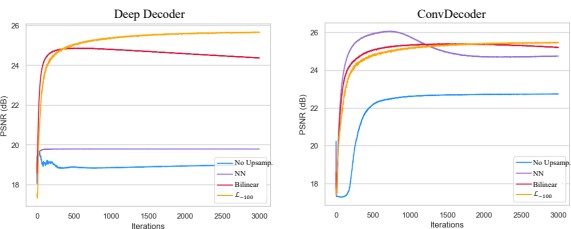

Figure 9: Results evaluated on the masked regions averaged across 30 slices. The unlearnt upsampler critically influences both the peak PSNR and the susceptibility to overfitting.

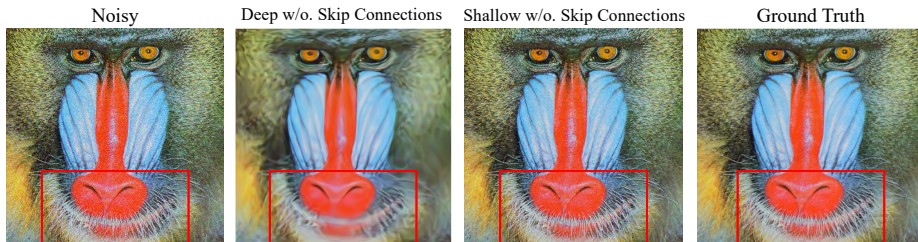

Figure 10: **Denoising** experiments. Deeper architectures with few or no skip connections tend to generate smoother outputs compared to the shallower ones.

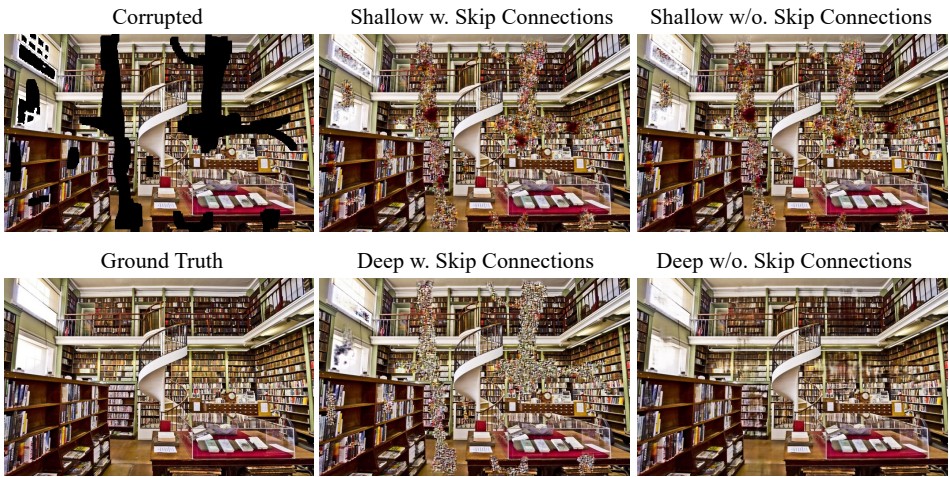

Figure 11: **Inpainting** experiments. Deeper architectures with few or no skip connections tend to generate smoother predictions for the masked regions than the shallower architectures. Skip connections make deep architectures perform similarly as the shallower ones.

### A.4.2 NATURAL IMAGE EXPERIMENTS

We reaffirmed our observations above on **image denoising**, which is a natural application of DIP. The results in Fig. 8 show a very similar trend as in MRI reconstruction. We further validated on a challenging **image inpainting task** that inherently resembles the case in MRI reconstruction. The results are shown in Fig. 11 and Fig. 12

We argue that the understanding about the upsampling and its interactions with other architecture elements can help explain why deeper networks with fewer skip connections converge more slowly, generate smoother outputs and are less prone to overfitting (Sec. 4). Concretely, the upsampling operation inserted in-between the decoder layer slows down the generation of high frequencies required for transforming the lower-resolution feature maps into the higher-resolution target image, primarily due to its role as a fixed low-pass filter. As the network depth increases, the degree of smoothness increases (Fig. 10). Skip connections notably accelerate the convergence (Fig. 12) and ameliorate the over-smoothing issue, likely due to the reduced "effective" upsampling rate.

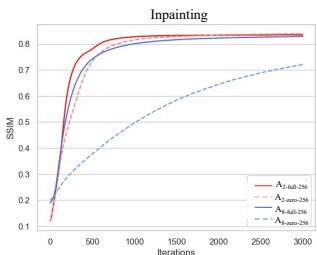

Figure 12: Deep architectures with zero skip connection converge more slowly, i.e., $A_{8-zero-256}$

All these observations are consistent with our MRI experiments in Sec. 4.

## A.5 EXAMPLE RESULTS ON $8\times$ UNDERSAMPLING

DIP with an inappropriately chosen architecture exhibits even more severe reconstruction artifacts in $8\times$ undersampling, which may not be remedied by early stopping as even the peak PSNR could be low (see metric curves). Nevertheless, our method substantially alleviates the artifacts while employing the same architecture. Particularly, we found that scaling up not only the sigma but also the kernel size of the Gaussian blur improves the visual quality in such a high undersampling rate.

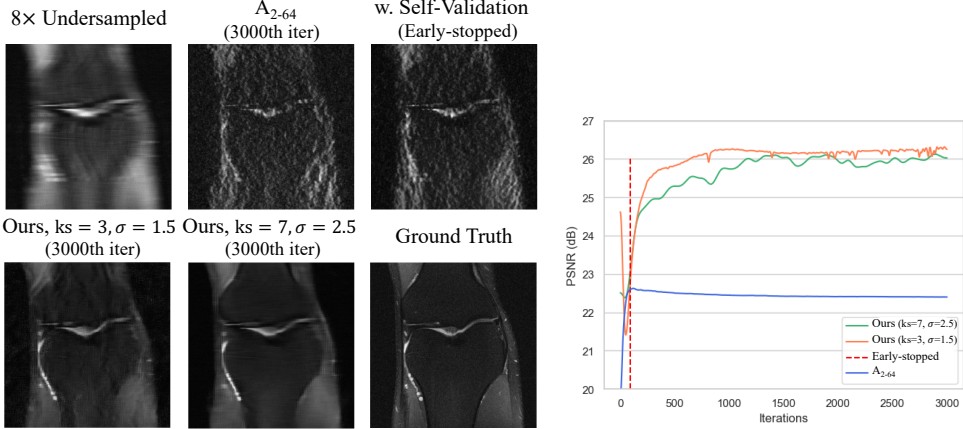

Figure 13: Qualitative results of $8\times$ undersampling. All methods were evaluated on $\mathbf{A_{2\text{-}64}}$.

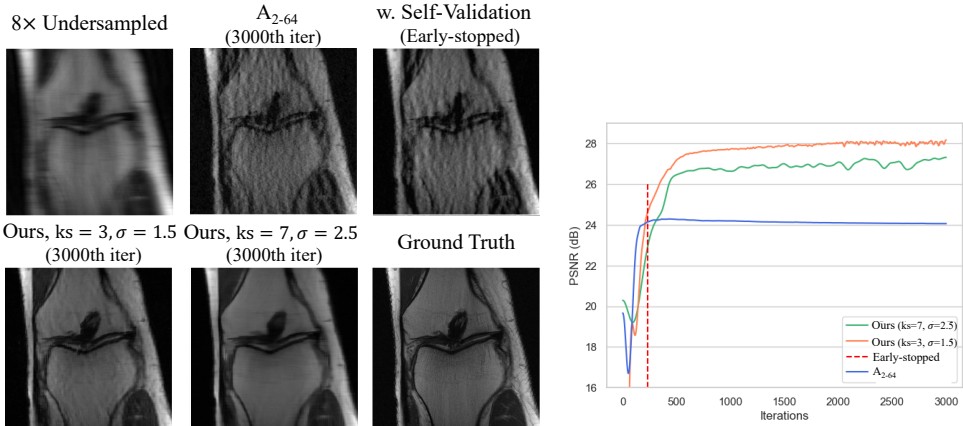

Figure 14: Qualitative results of $8\times$ undersampling. All methods were evaluated on $\mathbf{A_{2\text{-}64}}$.

## A.6 ANALYSIS ON SENSITIVITY TO HYPERPARAMETERS

As stated in the "implementation details" section, we set the filter size of the Gaussian blur to a fixed value, i.e., 3, and chose the sigma uniformly from a fixed range, i.e., $[0.5, 2.0]$. The substantially improved performance demonstrates that the method exhibits robustness to a certain range of hyperparameters. For the undersampling rate higher than $4\times$, a larger kernel size and sigma value are generally beneficial for better visual quality (see qualitative examples in Sec. A.5). We then test the sensitivity of the proposed Lipschitz regularization to its only hyperparameter - the regularization coefficient $\lambda$. The experiments were performed on the multi-coil knee validation dataset (Tab. 11).

## A.7 DETAILS OF THE CUSTOMIZED UPSAMPLER

The upsamplers experimented in Tab. 10 is constructed by first inserting zeros into the input (or feature maps) in an interleaved fashion, and then convolving with the filter with the following coefficients:

**Algorithm 1:** PyTorch-style pseudocode for customized upsampling

```python
# upx:  the upsampling scaling factor in the x direction
# upy:  the upsampling scaling factor in the y direction
# x:  the input to be upsampled
def InsertZeros(x, upx, upy, gain=1.0):
  b,c,h,w = x.size()
  x = x.reshape([b, c, h, 1, w, 1])
  x = F.pad(x, [0, upx - 1, 0, 0, 0, upy - 1])
  x = x.reshape([b, c, h * upx, w * upy])
  x = x * gain
  return x
# LPF construction
# w:  the coefficients
def lowpass_conv(num_chns, w, pad_size='same', pad_mode='zeros'):
  # filter size
  k_size = len(w)
  # Convert 1D LPF coefficients to 2D
  f_2d_coeff = torch.outer(w,w)
  f_weights = torch.broadcast_to(f_2d_coeff, [num_chns, 1, k_size,
   k_size])
  conv = nn.Conv2d(num_chns, num_chns, kernel_size=k_size,
   stride=1, padding=pad_size, padding_mode=pad_mode, bias=False,
   groups=num_chns)
  conv.weight.data = f_weights
  conv.weight.requires_grad = False
  return conv
```

Table 11: **Evaluation on hyperparameter sensitivity** of the Lipschitz regularization. PSNR values (↑) are reported. The chosen is underlined.

| Matrix norm | Hyper-param. | $A_{2-256}$ | $A_{2-64}$ | $A_{5-256}$ | $A_{5-64}$ | $A_{8-256}$ | $A_{8-64}$ |
|---|---|---|---|---|---|---|---|
| | $\lambda = 1$ | 28.41 | 29.21 | 29.17 | 29.79 | 29.43 | 30.14 |
| $\ell_\infty$ | $\lambda = 1.5$ | 27.89 | 28.98 | 28.68 | 30.11 | 29.13 | 29.42 |
| | $\lambda = 2$ | 28.36 | 29.25 | 28.51 | 29.60 | 28.98 | 29.52 |

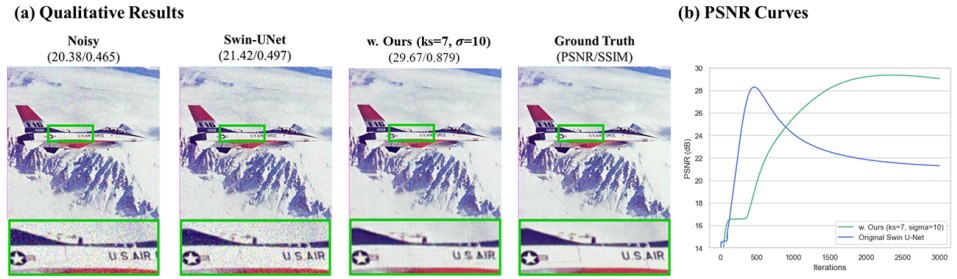

**(a) Qualitative Results**  **(b) PSNR Curves**

Figure 15: Example results of a transformer (i.e., Swin U-Net Cao et al. (2022)). The original Swin U-Net consists of only Swin Transformer blocks and skip connections, without upsampling layers. Our method substantially alleviates the overfitting and enhances the peak PSNR.

Nearest neighbor (NN): $[0.5, 0.5]$

Bilinear: $[0.25, 0.5, 0.25]$

$\mathcal{L}_{-100}$: $[0.000015, 0.000541, 0.003707, 0.014130, 0.037396, 0.075367, 0.121291, 0.159962, 0.175182, 0.159962, 0.121291, 0.075367, 0.037396, 0.014130, 0.003707, 0.000541, 0.000015]$

$\mathcal{L}_{-100}$ is designed using the Kaiser window, with the cutoff frequency as 0.1 and the Beta of the Kaiser window as 10.

Specifically, the customized filter can be constructed using the following code and can then be used as a plug-in module for any network architecture.

