# OpenReview forum: "Towards Architecture-Insensitive Untrained Network Priors for Accelerated MRI"
_ICLR.cc/2024/Conference — Submitted to ICLR 2024_

### Official Review · Reviewer_QL8v · 2023-10-16

**Soundness:** 3 good
**Presentation:** 3 good
**Contribution:** 2 fair
**Rating:** 6
**Confidence:** 5

**Summary:**

The paper tackles the overfitting issue of untrained networks from the perspective of non-architectural interventions. Specifically, the authors propose to (1) low-pass filter the network input, and/or (2) penalize the Lipschitz constant of the network to encourage more smoothness in the output and to prevent the network from overfitting.

**Strengths:**

The paper is overall well-written with a smooth flow.

The paper is filled with interesting and valuable experiments. The most prominent examples are Fig. 4, Fig. 6, and Tab. 4.

The idea of low-pass filtering the input noise with the goal of regularizing the smoothness-artifact trade-off in the output reconstruction is impressive. Likewise its effectiveness in lifting up the performance of under-performing models.

**Weaknesses:**

The problem has a rich literature, some of which are already cited by the authors. What is missing is comparison to those baselines. e.g., self-validation based early stopping is also a non-architectural (and popular) regularization to avoid overfitting. Thus, it’s essential to compare against that baseline (and potentially a few more). The authors have confined themselves to a mere disadvantages summary of such important baselines in the introduction.

Section 4 is presented as the first systematic study on architectural dependencies of untrained networks. However, it is redundant with the investigations already done in the literature. e.g., [1] Appendix B already discusses such design choices conclusively.
[1]Darestani, M.Z. and Heckel, R., 2021. Accelerated MRI with un-trained neural networks. IEEE Transactions on Computational Imaging, 7, pp.724-733.

A very interesting case to explore would be investigating architectures that still incur overfitting after applying the proposed regularizations; something not explored in the paper.

Minor

The space before “(“ is often omitted. e.g., many cases in  paragraphs 2&4 of page 3.

Since section 4.1 is also and experimental section in its nature, it’d be useful to have the dataset setup here (similar to section 4.2 or 6). This is because untrained networks require different architectures for each anatomy.

**Questions:**

- Fig. 3 claims deeper and narrower networks are less prone to overfitting. However, the metric to draw this conclusion isn’t fair. Instead of measuring the absolute masked SSIM curves, one should measure the slope of the fall and the convergence value since it isn’t surprising to see deeper networks overfitting at later stages during the course of optimization.

Furthermore, the claim tries to partly deliver the point that #parameters isn’t the primary factor in overfitting; however, the 2-layer 256-channel network has indeed 4x more parameters compared to the 8-layer 64-channel network according to Tab. 3, and therefore more prone to overfitting because of its #parameters?
Finally, is Fig. 3 averaged over multiple examples or only the results of one sample? If it’s just one sample, it’d be hard to draw such bold conclusions.

- What are the authors’ thoughts on using their method for 8x? The reviewer fully understands that given the limited rebuttal time, it’s not reasonable to ask for conclusive 8x experiments. But given the fact that the difference between untrained and trained networks enlarges by going from 4x to 8x acceleration, would it be possible to claim that the proposed regularization schemes may help reducing that gap?

Minor

- Isn’t there a better way to design Tab. 2? Because currently, the labels Depth, Width, Kernel size, and Skip connections are placed in a very confusing way.

---

> ### Author Response · Authors · 2023-11-22
> **Reply to reviewer QL8v (1/2)**
>
> We sincerely thank the reviewer for the thorough review, encouraging evaluation and insightful suggestions. We provide our responses below.
>
> **W1: Comparison to self-validation**
>
> Thank you for this suggestion! We have now incorporated the comparisons with self-validation in suppl. A.2. As demonstrated, our method outperforms self-validation, particularly on underperforming architectures. When combined, the accuracy surpassed that achieved by using self-validation alone.
> \
> This is because, while self-validation mitigates overfitting by stopping near the peak PSNR, it does not fundamentally address the issues of underperforming architectures, i.e., the peak PSNR remains subpar. Our method generally enhances the peak PSNR (Fig.7, 13, 14, 15) while alleviating overfitting - the latter can be seen in the relatively small variance in the performance of most evaluated architectures when our methods are applied, regardless of whether (self-validation) early stopping is employed or not.  Hence, our method can readily integrate self-validation to shorten the runtime while maintaining (or improving) accuracy. This also exemplifies that our method may potentially be combined with other advancements in untrained networks as well, because our focus on reducing the architectural sensitivity of untrained networks, to the best of our knowledge, is orthogonal to the existing works.
>
> **W2: Section 4 presented as the first systematic study, but investigations were already done in [1] Appendix B**
>
> Thank you for pointing this out. While we may arrive at a similar conclusion about the generally well-performing architectural characteristics as in [1], we believe that our investigations are unique in several ways:
>
> 1. We pinpointed the importance of unlearnt upsampling to the success of untrained networks (Sec. 4.1). This is further validated in suppl. A.4 where we showed that simply changing the bandwidth of the low-pass filter within the unlearnt upsampling substantially influences both the peak PSNR and the susceptibility to overfitting, and nearly closes the performance gap between two architectures with distinct network sizes.
>
>
> 2. The understanding of the upsampling allows us to infer the roles of other architectural elements, e.g., skip connections and network layers, by assessing their combined influences on the final outcome (Sec. 4.2).
>
> In other words, our investigations are not only for identifying the desired architectural characteristics but more importantly, for understanding why they are more desired, which critically motivates our methods. That said, we agree that a different form of architectural investigation has been performed and we have now removed the word "first" in the newly uploaded PDF.
>
> **Investigating architectures that still incur overfitting after applying the proposed regularizations**
>
> Thank you for this thoughtful comment. Indeed, our method does not eliminate overfitting but only alleviates it. This can be seen from the improvement on certain architectures (e.g., $A_{5-{256}}$) after applying (self-vadliation) early stopping (Tab. 7 in suppl.). The choice of hyperparameters, i.e., the regularization granularity, might also be relevant here.
>
> **The space before “(“ is often omitted; adding the dataset setup in Sec. 4.1**
>
> Thanks for pointing these out. They have now been corrected/revised in the uploaded version.

---

> ### Author Response · Authors · 2023-11-22
> **Reply to reviewer QL8v (2/2)**
>
> **Q1:Claims about deeper and narrower networks are less prone to overfitting. 2-layer-256-channel network has more parameters than 8-layer-64-channel so more prone to overfitting? Is Fig.3 averaged over multiple examples or just one sample?**
>
> These are excellent questions. We note that the number of network parameters itself may not sufficiently explain why the 8-layer-0-skip-256-channel network converges more slowly than the 8-layer-8skips-256-channel network (Fig.3 in main text, Fig.11,12 in suppl.A.4, assuming upsampling is present), and why only changing the attenuation extents of the low-pass filter in the upsampler makes a significant difference on the performance (Fig.3 in main text, Tab. 10, Fig.8,11,12 in suppl.A.4). Besides, when the upsampling is absent,  the reconstruction capability of the network is clearly hampered (Fig.2 in main text, Fig.8,9 in suppl.).  All of these modifications do not scale the network size. Also, a deeper network (with zero skip) tends to generate smoother outputs compared to a shallower one (Fig.10 in suppl.). This might seem counter-intuitive, but understandable if taking into account the attenuation effects of the unlearnt upsampling. The # of params is not completely irrelevant, though, and this can be seen that when the depth is the same (the attenuation effects of the upsampling are fixed), the smaller the width is, the less prone to overfitting the network seems to be (by comparing the slopes of the SSIM curves in Fig. 3), and that when there is no upsampling, the network behavior becomes more correlated with the # of params (Fig. 8 in suppl.), which fits our intuition.  Based on all these analyses, we find the architectural configurations matter more than the overall number of parameters for untrained networks.   The results in Fig.3 are averaged across 30 slices.
>
>
> **Q2: Use our method for 8x undersampling?**
>
> Thank you for this thoughtful comment and for your understanding. It is indeed interesting to see if our method helps in such cases where overfitting may occur more easily, and we have now included some example results in suppl. A.5.  The results are similar as in the 4x case and our method is able to outperform the baseline by a large margin. The only difference we observed is that the tuning of the hyperparameters seems to be tricker under such a high undersampling rate, unlike in the 4x case where a randomly chosen sigma value from a fixed range already gives high performance. Notably, although our method significantly alleviates the artifacts in this case, the output may lose some details (which is expected, due to the limited information within the highly-undersampled measurements). We expect the integration of certain supervised signals from external data to potentially compensate for such information loss, which we leave for future work.
>
> **Better way to design Tab. 2?**
>
> We apologize for the confusion caused. As we are currently still constrained within the 9-page limit, we may not be able to make substantial changes to it at this point, but we will be sure to revise the table to ease the interpretation once additional space is allowable in our final version.
>
> Lastly, we would like to thank you again for all your thoughtful comments, which really helped improve our work. We would be more than happy to discuss further.

---

### Official Review · Reviewer_uRkm · 2023-10-28

**Soundness:** 1 poor
**Presentation:** 3 good
**Contribution:** 2 fair
**Rating:** 3
**Confidence:** 4

**Summary:**

This paper introduces an enhanced version of the deep image prior for MRI reconstruction. The contributions include the proposal of an optimized architecture (specifying width, kernel size, etc.) tailored for a specific experimental setup. Notably, the approach incorporates input coordinates instead of relying solely on white Gaussian noises and integrates Lipschitz Regularization into the training loss. The experimental validation is conducted on the fastMRI dataset, focusing on a 4x acceleration factor scenario.

**Strengths:**

- Comprehensive investigation on the architecture of deep image prior.
- This paper is in general easy to follow (though with some unclear parts, see weakness below).

**Weaknesses:**

- Overclaim: The author claims that the proposed architecture demonstrates optimal performance, a claim primarily based on observations from a specific data setup. Such a conclusion lacks merit without theoretical justification or validation across diverse data setups. It remains unclear if the proposed architecture would yield similar results in other scenarios. Notably, the comparison against a fully-sampled ground truth, while foundational, is impractical in real-world applications.

- Lack of Novelty: The realm of MRI reconstruction networks has reached a saturation point in performance (see also the results shown on the fastMRI leaderboard). Therefore, any new approach must offer a distinct advantage absent in existing methods. While incorporating input coordinates and applying Lipschitz regularization might be novel in the context of DIP for MRI, these contributions might not significantly impact the broader MRI reconstruction community.

- Performance: The fastMRI 4x acceleration challenge is widely acknowledged as non-challenging for deep learning models. Reference to Table 12 in the fastMRI dataset paper (https://arxiv.org/pdf/1811.08839.pdf) indicates that classical TV methods, despite being non-learning-based, yield comparable results to those presented in this paper (both in terms of PSNR and SSIM). Furthermore, the UNet results outperform the proposed method by a significant margin.

**Questions:**

- Why does the author not compare against with end-to-end learning method, such as Unet or some deep unfolding baseline like VarNet?
- The Brain setup in Table 6 seems never been mentioned in this paper.

---

> ### Author Response · Authors · 2023-11-21
> **Reply to Reviewer uRkm (1/2)**
>
> Thank you for your detailed and constructive comments.  We believe there may have been a few misunderstandings, and we hope to address them below.
>
> **W1: Overclaimed that the proposed architecture demonstrates optimal performance, observed from a specific data setup.**
>
> We did not propose any new network architecture, nor did we claim that any specific architecture achieved “optimal” performance.  Instead, we introduced architecture-agnostic regularization methods and demonstrated their effectiveness in enhancing performance across various network configurations, with the ultimate goal of alleviating the need for architectural tuning. Our method stems from our analysis of the architectural influences of untrained networks. In fact, the characteristics of the well-performing architectures mentioned in the paper (if this is the “proposed architecture” to which the reviewer is referring) have already been widely adopted in various untrained inpainting-like reconstruction tasks, including MRI [1], CT [2] and natural images [3], as discussed in Sec. 4.2. We validated and showed that the counterparts performed worse indeed and analyzed the underlying reasons. \
> \
> We started by noting the importance of the unlearnt upsampling in DIP (Sec. 4.1), which, by nature, is composed of a fixed low-pass filter [4]. Given that the success of DIP has been widely attributed to network bias towards low frequencies [5], it is both reasonable and intriguing to investigate the extent to which an explicit low-pass filter within an architecture, such as an interpolation-based upsampler, contributes to the reconstruction performance. Our investigations reveal that upsampling critically influences both the peak PSNR and the susceptibility to overfitting (main text Fig.2, suppl.Tab.10, Fig.8,9). Then, by examining its interactions with the network layers and skip connections, we provided explanations on how different architectural characteristics influence the final outcome in Sec.4.2. We agree that more validations in this regard would be more helpful and we have now provided more evidence from the additional MRI experiments, image denoising and inpainting in suppl. A.4. \
> \
> Moreover, we stated in Sec.4.2 in our first submission that despite knowing the characteristics of generally better-performing architectures for inpainting-like tasks, it is still difficult to specify an “optimal” architecture for a given anatomy, and hence we presented our remedies that can generally mitigate the performance gap among different architectures, which were also inspired by our architectural investigations.
>
> **The comparison against the fully-sampled ground truth is impractical in real-world applications.**
>
> We agree, but as our focus is on improving untrained reconstruction rather than unsupervised quality control, we followed the standard of most current works [1,6] and did not add another layer of complexity to the evaluation at this point.
>
> **W2: Lack of novelty: the realm of MRI reconstruction networks has reached a saturation point in performance (shown on the fastMRI leaderboard); contributions not significantly impact the broader community**.
>
> We would like to clarify that although we used the fastMRI dataset for evaluation, our primary objective is not to introduce another new supervised network for benchmarking. Instead, our focus is on exploring and enhancing a well-known, yet poorly understood, unsupervised method (i.e., DIP) on a publicly available dataset such that future works can more easily build upon. In this regard, we strongly believe that our paper provides new perspectives and methods for untrained image reconstruction.
>
> **Reference** \
> [1] Darestani et al., Accelerated MRI with untrained neural networks. IEEE Transactions on Computational Imaging, 2021. \
> [2] Barbano et al.,  An educated warm start for deep imae prior-based micro ct reconstruction, IEEE Transactions on Computational Imaging, 2022. \
> [3] Ulyanov et al., Deep image prior, CVPR, 2018. \
> [4] Gonzales, et al., Digital image processing. Addison-Wesley Longman Publishing Co., inc., 1987. \
> [5] Shi et al., On Measuring and controlling the spectral bias of the deep image prior. IJCV, 2021. \
> [6] Yaman et al., Zero-shot self-supervised learning for MRI reconstruction. ICLR, 2022. \
> [7] Zbontar et al., fastMRI: An open dataset and benchmarks for accelerated MRI. arXiv, 2018.

---

> ### Author Response · Authors · 2023-11-21
> **Reply to Reviewer uRkm (2/2)**
>
> **W3: Performance comparable to TV**.
>
> This may be a miscommunication as according to Table 12 in the fastMRI paper [7], the PSNR and SSIM values are \
> \
> [**knee**] TV: 30.05 / 0.632, Ours: 31.93 / 0.776. [**brain**] TV: 27.53 / 0.444, Ours: 33.12 / 0.870.    (*Ours denotes the improved $A_{2-{64}}$)
>
>
> **Outperformed by U-Net**. \
> The untrained network is a single-instance reconstruction approach, while supervised methods are trained on a database. Hence, supervised methods may outperform untrained networks when the test data are within the training data distribution. However, any changes in the acquisition parameters such as the sampling rate or modality shift may result in a decay in performance for supervised methods, as systematically studied in [8].  On the other hand, untrained networks are agnostic to such changes as they are directly optimized on the test data, i.e., zero-shot.  Therefore, they are more advantageous on out-domain data, as shown in our results (suppl. A.3). It is also worth noting that our modifications significantly mitigate the performance gap between the untrained network and the trained U-Net on in-domain data. We believe that the gap can be further reduced through a thorough understanding of untrained networks.
>
> **Q1: Why not compare with an end-to-end learning method, such as U-Net?** \
> The primary focus of our paper is centered around improving our understanding and the performance of untrained networks. Upon the reviewer's request, we have now included a comparison in suppl. A.3.
>
> **Q2: Brain setup never mentioned**. \
> Thanks for pointing this out. We briefly mentioned in the Datasets section that we used AXT1 brain slices for evaluation in our first submission. We have now added more details under the 9-page limit.
>
>
> Finally, we would be glad to engage in discussion if there are any other comments/questions.
>
> **Reference** \
> [7] Zbontar et al., fastMRI: An open dataset and benchmarks for accelerated MRI. arXiv, 2018. \
> [8] Darestani et al., Test-time training can close the natural distribution shift performance gap in deep learning based compressed sensing. ICML, 2022.

---

> > ### Comment · Reviewer_uRkm · 2023-11-23
> > **Thanks for the rebuttal**
> >
> > I would like to thanks for the authors' efforts in this rebuttal. Some of my concerns have been addressed. Please see the remaining below:
> > - **W1**: The logic of this paper, from my understanding, is to try different architectures in a specific dataset (fastMRI) and problem (4x MRI reconstruction), and claim that some patterns have been found. Based on these patterns, the authors propose two regularizers for DIP. My points are: those patterns (i.e., low-pass) were only from a specific case; It is suspicious that if these **heuristic** observations/conclusions can be generalized to different datasets and/or problems. I think a solid/safe solution is to validate your observation in diverse datasets or to conclude your observation from a theoretical aspect.
> > - **W2**: Contributing on an unstudied topic does not always lead to an impact on the society. In terms of DIP, its suboptimal performance is well-known in MRI reconstruction (can also see from the authors' results compared to U-Net and poor visual results of the x8 reconstruction results). While the authors claimed that DIP had better generalization ability than U-Net, it is still doubtful if we would trade a huge performance drop for the better generalization ability. That being said, I still cannot see enough impact from this paper to the MRI reconstruction community.
> >
> > Moreover, this paper also compared against Self-Validation (Yaman et al., 2021), but Self-Val has much worse performance than what Yaman et al., 2021 reported (see Table 1 in Yaman et al., 2021). Why? My hypothesis is that Yaman et al., 2021 uses deep unfolding, a deeper CNN with forward model information embedded. It seems we could go back to W1: Are the authors's observations/conclusions transferable to the deep unfolding-based Self-Val?
> >
> > Given all of above, I will keep my original score.

---

### Official Review · Reviewer_uVud · 2023-10-30

**Soundness:** 3 good
**Presentation:** 3 good
**Contribution:** 3 good
**Rating:** 6
**Confidence:** 4

**Summary:**

This paper focuses on improving the reconstruction of Magnetic Resonance Imaging (MRI) data from under-sampled measurements. It introduces untrained networks inspired by the "deep image prior" concept, which relies on architecture rather than paired measurements. The study systematically analyzes the impact of architectural components on MRI reconstruction quality, leading to the identification of influential factors. The paper proposes architecture-agnostic remedies to mitigate overfitting in underperforming networks, enhancing reconstruction efficiency and robustness.

**Strengths:**

1) The paper conducts a systematic and comprehensive analysis of the architectural influences of "deep image prior" like methods on MRI reconstruction. It identifies key components that affect reconstruction quality, providing valuable insights for researchers and practitioners. To my knowledge, this has not been investigated before in the way that the paper does.
2) Motivated by their investigation, the authors propose architecture-agnostic remedies to mitigate overfitting in underperforming networks are practical and computationally efficient, offering a solution to enhance reconstruction efficiency without extensive architectural modifications.
3) The paper supports its findings with extensive experiments, demonstrating the effectiveness of the proposed methods.
4) The paper is well-structured and effectively communicates its methodology, findings, and contributions, making it accessible to a broad audience.

**Weaknesses:**

1) The study primarily focuses on MRI reconstruction, which is a specific application in medical imaging. "deep image prior" like methods are generic and I wonder what modifications, if any, are required to make the proposed solutions work for other domains.
2) The paper heavily emphasizes the architectural aspects of the problem, but it does not explore other potential factors that might affect MRI reconstruction quality, such as data acquisition protocols. Do the authors observe any different conclusions when the data undersampling factor is changed?
3) There is limited discussion on how the hyperparameters were selected? Are there any practical recommendations on how to select hyperparameters such as sigma of Gaussian blur etc.? Will it depend on the degree of under sampling in the data?

**Questions:**

Please see weaknesses above

---

> ### Author Response · Authors · 2023-11-22
> **Reply to reviewer uVud**
>
> We sincerely thank the reviewer for a careful review of our work, positive feedback and thoughtful comments.  We provide our responses below.
>
> **What modifications are required to make the proposed solutions work for other domains?**
>
> Thank you for this excellent question! For a better illustration, we have now incorporated experiments on denoising and inpainting for comparisons in suppl. A.5.  As demonstrated, the smoothing effects imposed by the analyzed architectural components on the outputs are quite consistent in these different tasks. The only thing is that different tasks might operate best under different levels of smoothing / spectral bias. A prominent example would be Fig.10 and Fig.11, where the same deep architecture enables better inpainting results by smoothly interpolating onto the masked regions, while producing over-smoothed output for a denoising case.  Since our regularization methods effectively promote smoothness in the output, they can be applied with adjustable degrees of smoothness where applicable.
>
> **Conclusions different when the data undersampling factor is changed?**
>
> Thanks for bringing up this important question. We have now experimented with the case of 8x undersampling and included the results in suppl. A.5. The results are similar as observed in the 4x case where our method is able to substantially alleviate the artifacts and improve the baseline model by a large margin. Considering the limited information within such highly undersampled measurements, we expect the integration of certain supervised signals from external data to potentially compensate for the information loss, which we leave for future work.
>
> **Practical recommendations on hyperparameters selections? Will it depend on the degree of undersampling in the data?**
>
> The reviewer is right that the hyperparameter indeed depends on the degree of undersampling. The only difference we observed in the 8x case is that the tuning of the hyperparameters becomes tricker, unlike the 4x scenario where a randomly chosen sigma value from
> a certain range (Sec. 6.2) already gives high performance. Particularly, we found that scaling up the kernel size is also helpful, in addition to the sigma value, under such a high undersampling rate.
>
> Lastly, we would like to thank you again for all your valuable comments, which greatly helped enhance the practical aspect of our method. We would be more than happy to clarify further.

---

### Official Review · Reviewer_a8Rv · 2023-11-01

**Soundness:** 2 fair
**Presentation:** 3 good
**Contribution:** 2 fair
**Rating:** 5
**Confidence:** 3

**Summary:**

This study investigates the performance of untrained neural networks for MRI reconstruction tasks. To address the unclear architecture design choice for achieving good reconstruction quality, the author come up with two observations:
1. The noise input with low frequency constraint helps to improve the reconstruction quality.
2. The introduction of a collection of learnable Lipschitz constant can fix the performance gap among different model architectures.

On the two MRI datasets, the paper shows that proposed approaches can stablize the reconstruction performance across models with different kind of hyper-parameters.

**Strengths:**

1. Investigating the untrained neural networks (Deep Image Prior) on MRI reconstructions is new and interesting.

2. The proposed approaches are simple and can be easily reproduced assume the baseline codes are accessible.

3. With such minor modifications, the new method outperforms the baseline DIP method.

**Weaknesses:**

1. Thought the idea of applying DIP to MRI reconstruction sounds interesting, I am worried about the potential impact this study can reach.
Considering the unclear trade-off between the additional optimization time per each image VS the one-time training budget, I don't think this idea is well motivated for medical image application. To address this, the authors could either present the table of estimated optimization runtime. Or the authors could show DIP-related methods clearly outperform the standard supervised methods.


2. It seems to me that the model hyper-parameter insights are derived from the baseline CNN model, it would really enhance the paper if the architecture study could have included detailed transformer or nas experiments, other than a short discussion presented in the introduction.

**Questions:**

Please see the above weakness section.

---

> ### Author Response · Authors · 2023-11-21
> **Reply to Reviewer a8Rv**
>
> Thank you for your thoughtful and constructive comments. We hope to address your concerns below. \
> \
> **W1: Unclear trade-off between the optimization time per image vs the one-time training budget; not well-motivated for medical image applications**.
>
> We thank the reviewer for the valuable suggestions. We have now included the runtime and accuracy comparisons in suppl. A.3.  While our work focuses on improving the understanding and accuracy of untrained networks, our method can readily integrate other advancements in untrained networks such as self-validation based early stopping (Tab.3 in suppl.) to significantly shorten the running time (~ 5mins/slice)  - conversely, our method improves the peak PSNR than when using self-validation alone.
>
> Supervised methods typically take days (>=3) for training, and may require re-training when the sampling pattern or modality changes. As also shown in the results, when directly applied to out-of-distribution data, supervised methods may suffer degradation in performance, while untrained networks are agnostic to such changes as they are directly optimized on the test data, i.e., zero-shot. We would like to emphasize that zero-shot learning is also important particularly in certain medical applications, where it is difficult or sometimes infeasible to acquire a large number of training data, such as free-breathing cardiac MR imaging and contrast-enhanced liver imaging at high spatiotemporal resolutions. Data collection for these applications can be time-consuming, costly and impractical. Additionally, rare pathologies may not have sufficient examples for training [1]. Another practical setting where longer reconstruction times are acceptable would be cases where image readings are done hours or days after the acquisition.
>
> We also kindly note that minor modifications like ours, allowing such a boost in performance, actually indicate that the potential of untrained networks has not been fully exploited. We believe that a more thorough understanding of untrained networks can further reduce the performance gap with the supervised methods and that our work could be one of the cornerstones.
>
> **W2: Only consider baseline CNN model; better if included transformer or NAS experiments**.
>
>
> Thank you for this insightful comment. Currently, NAS-DIP approaches are mostly developed for natural image applications [1], and the primary reason that motivates our architectural investigations is to avoid doing NAS for medical applications, as it is known to be computationally expensive and requires abundant training data with unclear generalizability. Notably, the architectural components considered in our work mostly overlap with those considered in NAS for DIP.  For example, NAS-DIP [2] mainly searches from kernel sizes, several common upsampling types and the patterns for cross-level skip connections. The latter significantly complicates their search space.  We argue that such a large search space may not be necessary, because the key components may not be that many, as evidenced in our experiments. Concretely, we showed that the unlearnt upsampling with a fixed interpolation kernel is the main driving force behind the success of DIP, and that such understanding helps identify the roles of other architectural properties such as skip connections in affecting the final outcome, as demonstrated in both MRI reconstruction and natural image experiments (Sec.4 in main text, suppl. A.4).  Compared to the black-box searching via NAS, we offered a quicker and more interpretable architectural design guideline for untrained networks, which might be more practical for medical applications.
>
> For the **transformer experiments**, due to the time limit, we have now included an example denoising result in **suppl. Fig.15**, where our methods also substantially alleviate the overfitting issue of a Swin transformer based U-Net (Swin-UNet) and enhance its peak PSNR. Although we can't possibly cover all sorts of architectures in one paper (our apologies), we believe that our work provides useful insights for future investigations.
>
> Thank you again for these insightful comments. Please let us know if we can clarify further.
>
> **Reference**\
> [1] Knoll et al., Assessment of the generalization of learned image reconstruction and the potential for transfer learning. Magnetic Resonance Medicine, 2019. \
> [2] Chen et al., NAS-DIP: Learning deep image prior with neural architecture search. ECCV, 2020. \
> [3] Cao et al., Swin-unet: Unet-like pure transformer for medical image segmentation. ECCV, 2022.

---

### Author Response · Authors · 2023-11-19
**General response to reviewers**

We would like to thank all the reviewers for their time and expert feedback.   We are encouraged that the reviewers think that the submission provides **valuable and interesting insights and experiments** (QL8v, uVud), **the idea and its effectiveness are impressive** (QL8v), the investigation and analysis are **comprehensive** (uVud, uRkm), **findings well-supported** (uVud), **well-written** (uVud, QL8v), and **the methods are practical** (uVud), **easily reproducible** (a8Rv),  **outperforming the previous method with such minor modifications** (a8Rv).

Upon the reviewers' requests and suggestions, we have now extended the supplementary materials to include the following experiments (marked in blue):
 \
\
&nbsp;&nbsp;&nbsp;1. Comparison to **self-validation (Section A.2)** [QL8v] . Particularly, we showed that our method can integrate with self-validation based early stopping to achieve better results with shortened runtime.  \
&nbsp;&nbsp; 2. Comparison to a **supervised method (Section A.3)**  in terms of performance and runtime [a8Rv, uRkm]. \
&nbsp;&nbsp; 3. Testing of **a different undersampling rate, e.g., 8x, (Section A.5)** [QL8v, uVud]. \
&nbsp;&nbsp; 4. To show the generalizability of our findings, we have added the **Section A.4 The "Devil" is in the Upsampling** with results demonstrated on more MRI experiments and broader domains such as image denoising and inpainting [uVud, uRkm,QL8v].
\
\
 We will also provide our point-to-point response to the individual comment.
\
\
With all these experiments, we hope that our submission can be seen as helpful for a better understanding of the working mechanisms of untrained networks and for unlocking its full potential as an unsupervised reconstruction method.

\
Thanks again,
\
the authors

---

### Meta-Review · Area_Chair_cf1A · 2023-12-17

**Metareview:**

The paper presents an approach to better understand deep image priors for MRI reconstruction, offering new insights and two architecture-agnostic strategies to narrow the gap between ill-designed models and well-performing ones. While the paper clearly has merits, reviews are
quite mixed (even after the rebuttal). Importantly, this would still be the case, even if one would consider some issues as resolved (which
the AC does, in light of the thorough rebuttal by the authors). However, in terms of being ready for publication, my opinion after reading the paper myself is that there are too many substantial changes (either in terms of writing or experiments) that would have to be made in order to push this across the acceptance threshold. Hence, I am recommending rejection at this point, knowing that this is frustrating, but the amount of work put in by the authors in the rebuttal period should make it relatively easy to address all raised concerns and re-write part of the paper for a re-submission.

**Justification For Why Not Higher Score:**

The current comments and concerns raised by the reviewers do not warrant an acceptance score at this time (i.e., Accept (poster)).

**Justification For Why Not Lower Score:**

N/A

---

### Decision · Program_Chairs · 2024-01-16

Reject